# Is Nigeria on course to achieve universal health coverage in the context of its epidemiological and financing transition? A knowledge, capacity and policy gap analysis (a qualitative study)

Yewande Kofoworola Ogundeji [ID] ,[1,2] Oluwabambi Tinuoye,[1] Ipchita Bharali,[3] Wenhui Mao [ID] ,[3] Kelechi Ohiri,[1] Osondu Ogbuoji,[3,4] Nneka Orji,[5] Gavin Yamey[3,4]

¹Health Strategy and Delivery Foundation, Abuja, Nigeria
²Community Health Science, University of Calgary, Calgary, Alberta, Canada
³Center for Policy Impact in Global Health, Duke Global Health Institute, Duke University, Durham, North Carolina, USA
⁴Margolis Center for Health Policy, Duke University, Durham, North Carolina, USA
⁵Ministry of Health, Federal Government of Nigeria, Abuja, Federal Capital Territory, Nigeria

**Correspondence to**
Dr Yewande Kofoworola Ogundeji;
yewande.ogundeji@hsdf.org.ng

## ABSTRACT

**Objectives** This study aimed to assess Nigeria's preparedness to finance and drive the universal health coverage (UHC) agenda within the context of changing health conditions and resource needs associated with the disease, demographic and funding transitions. Nigeria is undergoing transitions in the healthcare system that include a double burden of infectious and non-communicable diseases, and transition from concessional donor assistance towards domestic financing for health. These transitions will affect Nigeria's attainment of UHC.

**Design and setting** We conducted a qualitative study, including semistructured interviews with relevant stakeholders at national and subnational levels in Nigeria. Data from the interviews were analysed using thematic analysis.

**Participants** Our study involved 18 respondents from government ministries, departments, and agencies, development partners, civil society organisations and academia.

**Results** Capacity gaps identified by respondents included limited knowledge to implement health insurance schemes at subnational levels, poor information/data management to monitor progress towards UHC and limited communication and interagency collaboration between government agencies and ministries. Furthermore, participants in our study expressed those current policies driving major health reforms like the National Health Act (basic healthcare provision fund) appear adequate to support UHC advancement in theory, but policy implementation is a key challenge due to a lack of policy awareness, low government spending on health and poor evidence generation for information to support decisions.

**Conclusion** Our study found major gaps in knowledge and capacity for UHC advancement in the context of Nigeria's demographic, epidemiological and financing transitions. These included poor knowledge of demographic transitions, poor capacity for health insurance implementation at subnational levels, low government spending on health, poor policy implementation and poor communication and collaboration among stakeholders. To address these challenges, collaborative efforts are needed to bridge knowledge

## STRENGTHS AND LIMITATIONS OF THIS STUDY

⇒ The sample included in this study comprised government officials from both national and subnational levels, development partners and academia active involvement in health-related programmes in Nigeria.
⇒ Our study had a limited number of respondents from the subnational level, which means our findings may not be generalisable at subnational levels.
⇒ Our study respondents did not include policy makers or beneficiaries whose perspectives on capacity and policy gaps may differ.

gaps and increase policy awareness through targeted knowledge products, improved communication and interagency collaboration.

## INTRODUCTION

In line with the global commitment towards universal health coverage (UHC), Nigeria is making a push to reduce out-of-pocket (OOP) expenditure and improve the breadth of access and the quality of health services by increasing the population covered, services covered and proportion of costs covered.

The major challenges in the Nigerian health system are multifaceted and interconnected. OOP expenditure represents about 75% of total health spending in the country.[1 2] With less than 5% of the population having any form of health insurance coverage, there is a very high risk of impoverishment due to health expenses.[3] Nigeria's burden of reproductive, maternal, neonatal and child health conditions is among the highest in the world,[4] and it also has a high malaria and tuberculosis (TB) burden. Moreover, the country has an increasingly growing incidence of non-communicable diseases (NCDs) which have

been estimated to account for 29% of deaths in Nigeria in 2016.[5]

Despite considerable efforts, progress towards UHC targets in Nigeria has been slow.[6] The low levels of public health financing and high reliance on external support are concerning given the impending donor transitions and reductions in concessional external financing for health.[7] This delayed progress towards UHC is further threatened by a set of four interlinked health transitions: shifts in demography, disease burden, development assistance for health and domestic health finance (the '4Ds' of transition).[8] The shift in disease burden includes a growing incidence of NCDs and injuries, coupled with a high pre-existing burden of maternal, newborn and child health conditions and infectious diseases. The changing demography includes a bulge in the adolescent band of the population pyramid.

Achieving UHC within the context of these highlighted transitions requires intentional efforts and collaborations between multiple stakeholders within the health ecosystem. In addition, the extent to which UHC can be achieved and sustained is dependent on the knowledge and capacity of these stakeholders, as well as the availability and implementation of relevant policies that consider these transitions in the pursuit of UHC.

This study aimed to assess Nigeria's preparedness to finance and drive the UHC agenda within the context of changing health conditions and resource needs associated with the disease, demographic and funding transitions. We aimed to identify and describe gaps in knowledge, capacity and policy of UHC financing and advancement within the context of these health transitions that Nigeria is experiencing. Our study presents new and contextual information and pragmatic recommendations that are targeted at policy makers, implementers and other stakeholders to help strengthen Nigeria's readiness and the mechanisms for meeting the health needs of the population to achieve UHC.

## METHODS
### Study design
This study used a qualitative approach (in-depth interviews were conducted with key informants) to explore the current knowledge, capacity and policy gaps preventing progress towards UHC in the context of the disease, demographic and funding transitions, as well as opportunities, to close these gaps. The study included stakeholders from federal and state-level ministries, departments and agencies (MDA), development partner organisations, civil society organisations (CSO) and academia. Study findings from the key informant interviews were triangulated and validated with published health policy documents.

### Sampling
We used a type of purposive sampling known as 'maximum variation', whereby a small number of cases were selected such that diversity relevant to the research question is

maximised.[9] Study respondents were purposively selected based on their experience, position, capacity and active involvement in health-related programmes in Nigeria. Furthermore, a snowballing sampling approach was used where, following interviews, respondents were asked to recommend one or two persons in an ideal position to also participate in the study.[9] This approach ensured that perspectives from a wide range of stakeholder groups were captured to ensure variability and representation of diverse perspectives.

### Data collection
Key informant interviews were completed with 18 stakeholders. Participants were approached by the principal investigator via email. A topic guide (see online supplemental file 1) that was developed iteratively by the authors and in English was used to guide data collection. This topic guide was piloted with three respondents, and further refined to ensure that questions addressed the study objectives and were easily understood. Prompts and probes were used as needed, and respondents were allowed to share other insights on the health transitions in Nigeria that may not have been covered by the interview guide to allow for other emergent themes.

All interviews were conducted face to face (or virtually), in English and audio recorded with the consent of respondents. Interviews varied in length with an average of 45 min. The interviews were completed by a female research analyst trained in qualitative research methods (OT) who had no prior relationship with respondents. The interviewer took extensive notes during and after the interviews to support the analytical process.

Data saturation was reached after 12 interviews, and we stopped further snowballing at this point. However, we continued to interview additional respondents (previously identified before saturation) for validation, consistency of results and to further enhance the development of themes. No participants dropped out of the study.

### Patient and public involvement
There was no involvement from patients or members of the public in the design, or conduct, or reporting, or dissemination plans of the research.

### Data management and analysis
Interview data were analysed using the framework method for qualitative data analysis by Ritchie and Spencer.[10] This approach supported the identification of similarities and differences in the data as well as relationships between different parts of the data to aid the drawing of descriptive and/or explanatory conclusions grouped by themes. Data analysis was completed by OT, IB and YKO. Analysis began by reading transcripts and field notes multiple times to identify preliminary codes within initial themes, which were refined through multiple iterations between the researchers. Data from the transcripts were coded independently by at least two coders with quality assurance provided by the principal investigator to enhance

**Table 1** Description of study participants

| Stakeholder category | Respondents (n) |
| --- | --- |
| National-level government officials | 6 |
| Subnational-level government officials | 5 |
| Development partners | 5 |
| Academia | 2 |
| Total | 18 |

study credibility and validity. After coding, data were imputed into a framework matrix to identify patterns and connections between themes and participants. Data management and analysis were facilitated through NVivo V.11.

After analysis was completed, a summary of the findings was shared with study participants for validation and feedback.

## RESULTS

### Participants

We interviewed 18 respondents in total. Six respondents were officials of federal-level MDA, five respondents were from the subnational level. The government officials constituted six senior-level officials (including deputy directors and directors) and five technical officers (mid-level officials). We also interviewed five respondents from development/implementing partners and two representatives from CSOs and academia (table 1). No respondents withdrew from the study.

### Themes

We identified four themes centred around respondents' knowledge of ongoing transitions (disease, demographic, financing), their perspectives on UHC-related policy and financing in Nigeria, stakeholders' capacity for policy implementation and opportunities to improve low government spending on health. We have presented the themes and subthemes with illustrative quotes.

### Theme 1: stakeholders' knowledge and perspectives on Nigeria's disease, demographic and financing transitions

Most participants were able to demonstrate relevant knowledge of the financing of UHC within the context of the ongoing transitions. When asked about these transitions and their potential impact on the Nigerian health system, the most commonly expressed perspectives included knowledge of *disease burden, dwindling donor funds and limited fiscal space for health*. Demographic transitions did not appear to be a pertinent issue among respondents.

When asked about disease burden, many participants described the situation as a double disease burden as opposed to a shift in burden from infectious to NCDs, pointing out that Nigeria is still tackling infectious

**Table 2** Stakeholders' knowledge and perspectives on Nigeria's disease, demographic and financing transitions (summary of participants' quotes)

| Subthemes | Quotes |
| --- | --- |
| Double disease burden | 1. '*It's not a changing disease burden, instead of a single barrel, it's more of a double-barrel disease burden…*' (Federal-level stakeholder) 2. '*It's apparent that there's a shift in the disease, not a shift rather, it's more of a doubling of the disease burden.*' (?) |
| Dwindling donor funds | 3. '*…I mean the donors are, some of them are leaving, you know, the funding landscape is dwindling.*' (Academic stakeholder) 4. '*…from time memorial, maybe let's say a decade ago, the support from donor assistance has increased in the health sector so it's not dwindling, it has been increasing…*' (Federal-level stakeholder) |
| Limited fiscal space for health | 5. '*…well, so the challenge, to me, the number one challenge, I see here, is the fiscal space, I don't know if you understand what I mean by fiscal space, the fiscal space for spending, on expanding on universal health coverage is a huge challenge…*' (Development partner) 6. '*…we're going to be stupid as the federal government to keep increasing health budget because I don't think they [federal government] have the fiscal space to do that.*' (Federal-level stakeholder) |

diseases while NCDs are also coming to the fore (table 2, quotes 1 and 2).

Most respondents also expressed that donor funds are dwindling, with donors exiting and closing out programmes they previously sponsored. However, one participant in the finance sector at the federal level expressed a differing view that donor funds are increasing in other areas and not dwindling as is generally thought (table 2, quotes 3 and 4).

Some study participants expressed that the limited fiscal space for health is a key challenge to achieving UHC in the face of dwindling donor funds. In particular, respondents explained that Nigeria's macroeconomic outlook has been challenged because of the global decline in oil prices and that it is difficult to see in the foreseeable future how spending on health can be improved considering these challenges unless radical economic reforms are implemented (table 2, quotes 5 and 6).

### Theme 2: knowledge and capacity gaps impacting UHC advancement

Participants in our study highlighted several capacity limitations that would impact progress towards UHC. These capacity gaps include (1) poor information/data management to monitor progress towards UHC, (2) insufficient evidence generation for information to make decisions, (3) poor capacity for health insurance

**Table 3** Knowledge and capacity gaps impacting UHC advancement (summary of participants' quotes)

| Subthemes | Quotes |
|---|---|
| Poor information/data management to monitor progress towards UHC | 1. '*And then of course another thing that we have to look at is the documentation of even the disease burden. The magnitude of the kind of ailments and illnesses that we have and how that goes into getting data that would be effective for planning purposes and decision-making purposes.*' (Federal-level stakeholder)<br>2. '*…We don't have information and when we have information, do we have adequate capacity to translate this information to action? No.*' (CSO) |
| Poor knowledge and capacity on health insurance implementation at subnational levels | 3. '*…when our focus is UHC and investments on health insurance, what we would like to see in the nearest future is, more government supporting health insurance decentralization policy in Nigeria, where the states can develop and nurture and effectively manage their own state health insurance scheme.*' (Federal-level stakeholder)<br>4. '*I've met a couple of [policy makers] that had the mindset that health insurance is going to absolve [the] government of its financial responsibility in the health sector. [They seem to think] even health insurance will even mobilize money for [the] government to spend.*' (Development partner) |
| Limited knowledge and capacity on health financing and economics for UHC | 5. '*…to start looking at more capacity building, including training and hiring some people with background in economics or health financing, to deploy them to some of the planning departments at the federal [level] or even some of the agencies.*' (Development partner)<br>6. '*Financing. Understanding the economic analyses of any intervention, right? There's only so much money the country will always have.*' (Federal-level stakeholder) |
| Limited communication and interagency collaboration between MDAs | 7. '*…in Nigeria, most of the organizations have worked solo [alone] in whatever they do so, that interagency collaboration, it's actually weak, and seems troublesome in Nigeria simply because they've just not been happening.*' (Federal-level stakeholder)<br>8. '*…so, in retrospect, part of what I'm looking at is that probably going forward in terms of doing collaborations very well maybe probably we should foster more of effective collaboration, effective communication sort of, then let the roles be very very clear in terms of areas of interdependency…*' (Development partner) |

CSO, civil society organisation; MDA, ministries, departments and agencies; UHC, universal health coverage.

implementation at the subnational level, (4) inadequate training of stakeholders within relevant ministries on health financing and economics for UHC, (5) poor human resource capacity and infrastructure for service delivery and (6) limited communication and interagency collaboration between MDAs. Respondents in our study perceived that addressing these capacity gaps would go a long way in accelerating Nigeria's progress towards UHC.

*Poor information/data management to monitor progress towards UHC*
Participants expressed their concerns about weak data systems and lack of access to information for decision-making. Respondents emphasised that where information is available, it is limited and incomplete. They further explained that without data to measure progress, and to indicate and benchmark targets, it will be difficult to assess where Nigeria stands and what needs to be done to get to the desired point regarding the health transitions. Some participants also described limitations with the translation of available information or data into useful metrics that policy makers can understand to effect actions and/or make key decisions. These views are reflected in table 3, quotes 1 and 2.

*Poor knowledge and capacity on health insurance implementation at subnational levels*
Some participants, particularly federal-level stakeholders and development partners, argued that Nigeria

has low implementation capacity for state health insurance schemes, which is one of the key ongoing health financing reforms in the country. Participants perceived that stakeholders at the subnational levels did not have adequate capacity to design or manage efficient health insurance schemes and that this could hinder success in the mid to long term. A few participants also expressed concerns that many health stakeholders including policy makers viewed health insurance as some sort of a magic bullet without fully understanding its requirements and potential challenges that need to be addressed. These views are illustrated in table 3, quotes 3 and 4.

*Limited knowledge and capacity on health financing and economics for UHC*
Many study respondents identified potential opportunities to build the capacity of relevant stakeholders within government ministries and agencies at both the national and subnational levels. When study participants were asked what areas of capacity could be improved for UHC attainment, some stated knowledge of health financing and economics for a better understanding of how to attain UHC with limited resources. The opportunity for training of stakeholders within relevant ministries on health financing and economics for UHC was commonly expressed by federal-level respondents and development partners who felt that there was the need

for an understanding of health financing and economics to support programme design, implementation and resource management (table 3, quotes 4–6).

### Limited communication and interagency collaboration between MDAs

When respondents were asked about their perception of interinstitutional relationships, limited communication and poor interagency collaboration were frequently cited as challenges that could potentially be improved on. Participants explained that poor collaboration stems from poorly defined roles of agencies outside the health sector towards UHC attainment. Participants also stressed that if communication is improved and agencies begin to work in synergy, collaborating efficiently for UHC advancement with clearly defined roles and accountability assignments, UHC advancement may be propelled. Table 3 quotes 7 and 8 illustrate these perspectives.

### Theme 3: poor implementation of UHC-relevant policies in Nigeria

Overall, most respondents expressed that Nigeria has relevant laws and policies geared towards UHC attainment, and they did not perceive any major gaps in current policies. However, participants often cited that poor policy implementation was the main challenge. Most respondents emphasised a lack of policy awareness, low government spending on health, limited partner coordination and poor domestic resource mobilisation as factors that contribute to poor policy implementation.

### The potential and limitation of the basic healthcare provision fund in accelerating UHC in Nigeria

There was some consensus among participants that Nigeria had relevant laws and policies that could support the progress of UHC and the most commonly cited policy was the basic healthcare provision fund (BHCPF).[11] The BHCPF is a key component of the National Health Act signed into law in 2014. The Act, through the BHCPF, provides a legal framework to provide minimum basic healthcare to the poorest and most vulnerable population across Nigeria through health insurance schemes and decentralised facility financing. The BHCPF is primarily financed by 1% of the national consolidated revenue fund (CRF).[11] Participants described that the BHCPF was a good way to advance UHC if properly implemented because it creates additional fiscal space for health and opportunities for better resource pooling and efficiency. However, despite all participants referencing the BHCPF, most participants believed that the implementation of existing policies including the BHCPF was poor and limited. In particular, respondents in our study stressed that it was not enough as a country to have good policies that are well intentioned—it is equally important to deliver on them through effective and timely implementation. Another participant explained that BHCPF implementation was slow even though the policy is expected to have a catalytic effect on healthcare delivery. Table 4 quotes 1–3 illustrate these views. Participants further

described factors that contribute to poor policy implementation as described in subsequent sections.

### Limited policy awareness across stakeholder groups

The lack of policy awareness was expressed by some respondents, and they believed that this lack of awareness cut across all stakeholder types, ranging from decision-makers to implementers, as well as citizens. Participants within academia, CSOs and development partners expressed that many key actors are not aware of policies that they ought to drive or participate in implementing and hardly speak about them. For example, one participant from civil society said:

> When you go to the local government who are supposed to implement at the primary healthcare level, all the principles we're talking about of universal healthcare, local resources for health, when you talk to them, you find out that they have no knowledge of what we're talking about. So, if the people and the organizations and the bodies and the agencies who are supposed to implement the decisions we're making at the national level, do not know what we're talking about, then what are we doing?

Participants further explained that policies are not properly implemented because of institutional capacity gaps. As a result of transfers within the civil service system, people who have been trained previously are sent to another MDA thus creating a knowledge and skills void in their previous position. Finally, participants further suggested the inclusion of communities in policy formulation to increase policy awareness. A few participants expressed that communities as recipients of the dividends of policies should be included at the formulation stage. They expressed those policies are often formulated centrally and expected to permeate communities that have limited knowledge of them. Participant quotes reflecting these views are seen in table 4, quotes 4–7.

### Limited partner coordination

Many respondents expressed that limited coordination among development partners for alignment was a key challenge and it was generally considered one of the major drivers of poor policy implementation. Study respondents believed that development partners ought to plug into existing government policies and programmes at national and subnational levels rather than introducing new programmes to suit their agendas. They emphasised that coordination between donors and partners will encourage alignment, promote synergy between the government's plans and those of donors and partners, avoid duplication of efforts and improve efficiency. Participants in our study also believed that this behaviour by development partners was primarily driven by the need for attribution because some donors and partners were interested in sole credit for contributions made and activities completed in line with their designed accountability frameworks. Table 4 quotes 8 and 9 illustrate these views.

**Table 4** Poor implementation of UHC-relevant policies in Nigeria (summary of participants' quotes)

| Subthemes | Quotes |
|---|---|
| The potential and limitation of the basic healthcare provision fund in accelerating UHC in Nigeria | 1. '*The basic health care [Basic healthcare provision fund], if started and if successful, I can assure you that it's going to be a very remarkable intervention that will help in the provision of the universal health coverage we are targeting. I have never seen any promising intervention like it and we're praying to see that it succeeds.*' (Subnational-level stakeholder)<br>2. '*BHCPF comes in to get more money into the system but also, to use it as a platform for better pooling and particularly pooling for primary health care…*' (Development partner)<br>3. '*What has always been the challenge is having the ability to actually implement these various plans and strategies.*' (Federal-level stakeholder) |
| Limited policy awareness across stakeholder groups | 4. '*I've never heard about the new [policy maker] talking about UHC, that he is making it a policy agenda. So that's part of the problem, we need drivers who are driving it. No driver, so you know if there's no driver within the health sector to drive these things, to get other people on board, create a multisectoral platform, and then you know make it into a movement otherwise we won't get there.*' (Academic stakeholder)<br>5. '*Then the social health insurance scheme too, awareness is just too low. People need to know that we still have to pay for health. Because of the way health has been introduced in the past, [people believe] it has to be free free free free, people don't seem to know that they have to pay for their health care, for the health care delivery services.*' (Subnational-level stakeholder)<br>6. '*Number two is the institutional capacity to implement UHC related policy framework. So, there's a huge challenge, you have in some ministry, inadequate manpower, very few people that are eligible, working in these areas. When you set up [a] health insurance scheme, to even get [the] right manpower to fill the position is a huge challenge, so, institutional capacity is a very big problem*.' (Development partner)<br>7. '*Many times, the people developing our policies are only looking at things from a facility, primary healthcare center perspective. We now need to start looking at the community perspective. We need to look at the people, not just at the services we are providing. We need to take our cultural and religious factors into consideration to make policies. They [Policy developers] do not understand that Nigeria is a complex country. Some of the things that work in Lagos, certainly may not work in Zamfara.*' (CSO) |
| Limited partner coordination | 8. '*…there's no overall organizing platform and authority, and that's where we took this bold approach to strengthen the federal ministry of health DPRS [department of planning, research, and statistics] to be coordinator and steward of the sector and to an extent, that has been lacking. It's causing [a] sort of fragmentation and might undermine things towards UHC.*' (Development partner)<br>9. '*We are not able to find or set an agenda for ourselves, you know, using our context-specific data to drive our program intervention so you end up seeing us paddling to whatever is the global agenda or what some really large donor comes and says this is what they want to focus on.*' (Federal-level stakeholder) |

BHCPF, basic healthcare provision fund; CSO, civil society organisation; UHC, universal health coverage.

### Theme 4: opportunities to improve low government spending on health

Nearly all participants expressed that government spending on health is low and the inability of government at federal and state levels to fund health programmes with meagre budgets. This was emphasised as a key challenge for UHC advancement and limitation for policy implementation in Nigeria. To address low government spending on health, respondents identified and described potential opportunities including domestic resource mobilisation, improved accountability for funds expenditure, multisectoral collaboration for health and prioritisation of health spending at subnational levels to allow subnational governments to take on additional financial responsibility for healthcare.

#### Low government commitment to health spending

Respondents expressed that subnational governments were not allocating enough budget to health, with spending on health being less than 3% of the total budget in some states. Respondents also argued that as a result of low commitment to spending on health, the delivery of many healthcare services is funded by donors. This was especially the case across major disease areas such as HIV, TB and malaria. For example, respondents emphasised that almost every TB medication consumed in the country was donor funded. Furthermore, participants expressed that government spending needed to improve given that donor funding is dwindling, and some health programmes previously funded by donors have been handed over to the government and that it was imperative that the government increase its spending commitment on health (table 5, quotes 1–3).

#### Accountability and multisectoral collaboration

Although participants expressed domestic resource mobilisation as an opportunity to address financing challenges, a few expressed that limited fiscal space may be a constraint. These participants further expressed the need for improved accountability in the expenditure of funds

**Table 5** Opportunities to improve low government spending on health (summary of participants' quotes)

| Subthemes | Quotes |
|---|---|
| Low government commitment to health spending | 1. '*The percentage of 15% budget for health from each state that was supposed to be put in the health sector, those percentages are not up to, it's not up to 3%. Some states find it difficult to even get 5% of their budget to be channeled to health or thereabout.*' (State-level stakeholder)<br>2. '*We just analyzed expenditure for HIV, TB, and malaria recently and it will shock you that in 2018, it was 2018 or 19. No, I think it was 18, 2018, on expenditure on tuberculosis, every TB drug that was swallowed in the country came from the donors. Not one came from the government of Nigeria, not [the] federal government, not [the] state government.*' (Federal-level stakeholder)<br>3. '*Let's take the HIV for example, we've known for years now that it [donor funding] is going to reduce [and] it's not sustainable. They [donors] are not going to continue to do everything for Nigeria but Nigeria still doesn't budget adequately for these services.*' (Federal-level stakeholder) |
| Accountability and multisectoral collaboration | 4. '*Not just putting money into health, but also accounting for how it is used and the result and performance out of the use of that. That conversation is very, very mute, and therefore with even more money going into health, and less accountability and documentation from revised outcomes of use for funding for health, we still have a problem with relating resources, financials, with the outcomes and how those are then driving our efforts to UHC.*' (Development partner)<br>5. '*You need resources. So, I mean, we are just talking about the health system actors, but you know UHC has more people outside the [health] system like [the] ministry of finance, the presidency, national assembly, you know so they need to work more, it needs a multisectoral action.*' (Academic stakeholder) |
| Prioritisation of health spending at subnational levels | 6. '*I think part of our problem too is that you know the states, most times they keep looking up to the federal to do something [fund]*.' (Federal-level stakeholder)<br>7. '*So, for me, if we want to achieve universal health coverage, we have to restructure the health system. We have to decentralize; we have to stop all these Abuja business*.' (Federal-level stakeholder) |

TB, tuberculosis; UHC, universal health coverage.

given the limited fiscal space. Respondents explained that it was important to connect expenditure to documented programme outcomes, and that doing this may influence improvements in government spending. In addition, some respondents expressed that interinstitutional collaboration between relevant MDAs was necessary to promote the UHC agenda. In particular, participants stressed the need for a stronger relationship between the ministries of health and finance to promote accountability and better value for money (table 5, quotes 4 and 5).

*Prioritisation of health spending at subnational levels*
Several participants believed that the decentralisation of healthcare to states and local governments was an opportunity that will propel them to take more responsibility. Decentralisation in this regard was particularly expressed relative to the dependence of subnational-level governments on the federal government for most of their health funds. Respondents said that often subnational tiers of government were too reliant on the federal level for funding and responsibility for healthcare.

## DISCUSSION
This study aimed to identify and describe gaps in knowledge, capacity and policy of UHC financing in Nigeria within the context of demographic, disease and funding transitions in the health sector. Participants in our study identified several capacity gaps. These include: (1) poor information/data management to monitor progress

towards UHC, (2) limited evidence and information to make decisions, (3) poor capacity for implementing health financing reforms at subnational levels and (4) limited communication and interagency collaboration. The most cited opportunity to improve capacity was in communication and interagency collaboration. Participants believed that if communication improved and stakeholders across government ministries worked more cohesively, there would be progress towards attaining UHC. These findings align with well-documented evidence that reflects the importance of communication and collaboration between key actors to promote synergistic action to drive progress towards UHC, especially in resource-constrained settings.[12]

Generally, respondents highlighted several existing policies that may support ongoing transitions, but they emphasised poor implementation of these policies. Participants elaborated particularly on the BHCPF as a viable policy most likely because it is a major health reform currently being implemented in Nigeria at the time of the study. As such, participants considered the BHCPF a strong policy reform and believed that if properly implemented, it would bring Nigeria closer to attaining UHC. While respondents in our study considered the pace of BHCPF implementation slow and limited, recent changes reflect some considerations for ongoing health transitions. For example, recently, the guideline for implementation of the BHCPF was revised to ensure that the care package now covers screening and referral for diabetes

mellitus, hypertension and some other NCDs and NCD risk factors at primary health facilities.[13] The package designed to be delivered at the secondary level caters to the management of diabetes and hypertension, sickle cell disease and treatment of cardiovascular conditions, renal diseases and liver diseases.[13]

Although the guidelines for the implementation of the BHCPF now accommodate some NCDs in care packages, management of cancers is not included, which contributes significantly to the burden of NCDs in Nigeria.[5] In addition, there is no up-to-date evidence or data on NCDs which is consistent with our study findings on the lack of data and evidence to make key decisions. Consequently, the burden of NCDs in Nigeria remains uncertain which may present an implementation challenge in the near future. The BHCPF is funded by at least 1% of the CRF of the federation, which is a good step towards increasing government spending, it should be noted that overall government spending on health still significantly falls short of recommended benchmarks. In addition, the BHCPF operates through health insurance and direct facility financing mechanisms, which require strong financial management and accountability systems to succeed. However, there is evidence that the design and implementation of health financing schemes and direct facility financing need to be strengthened in Nigeria, especially at the subnational levels.[14 15]

Furthermore, respondents mostly described their perspectives with respect to disease and financing (domestic and donor) transitions demographic transition did not appear to be a priority. This could be for a number of reasons including limited knowledge and the effects of the current demographic transition (increased young adult population) in Nigeria was considered a less urgent issue compared with the other health transitions.

Another key finding was low government spending on health which was highlighted as a key contributor to poor policy implementation. Health budgets have remained low over the last 10 years. For example, federal budget allocations to health in the last 5 years fell from just 5.8% in 2015 to 4.2% in 2020.[16] Many respondents attributed the low government spending on health to limited fiscal space and lack of government prioritisation, which have been documented in several studies.[17 18] Increasing government health spending and financial protection for citizens remain a key priority. This can be facilitated through analysis and documentation of the links between government health investments to health outcomes and better communication with the Ministry of Finance. However, given the nature of the resource constraints, other viable mechanisms should be explored, including efficiency in resource allocation and use, as well as improved collaboration with other health sector-sensitive ministries including education, nutrition and environment, recognising their role in advancing UHC.

Participants further expressed that there is poor policy awareness among stakeholders. This might be because of limited capacity for policy implementation at the subnational level and a disconnect between policymaking and implementation relative to knowledge and translation. Therefore, there is a need to prioritise building capacity (as opposed to capacity filling) at the subnational level focused on evidence generation, knowledge translation and advocacy at the community level. Policy champions can also be used as a tool to promote policy awareness and inclusion of communities in policy formulation.

### Study limitations

A limitation of our study is the limited number of respondents from the subnational level. While their responses provided some state-specific insights, our findings may not be generalisable at subnational levels. In addition, our study respondents did not include policy makers or beneficiaries whose perspectives on capacity and policy gaps may differ.

### CONCLUSION

Our findings have important implications for accelerating UHC within the context of disease, demographic and health funding transitions in Nigeria. Insights from our study highlight knowledge and capacity gaps and several opportunities to strengthen capacity of key stakeholders in implementation of health financing reforms, evidence generation and knowledge translation. To address these capacity gaps, there need to be sustainable learning activities for stakeholders, which can be achieved by leveraging academic or research institutions at the subnational levels to ensure that training capacity is local, readily available and cost-effective. However, there is a need for additional research to explore capacity gaps in contributory health insurance and direct facility financing at the subnational level.

Participants in our study generally described the BHCPF as a viable health reform that can support UHC goals within the context of disease, demographic and funding transitions. However, limited coordination and collaboration among key stakeholders, poor policy awareness, low government spending on health and limited partner coordination were generally identified as key challenges limiting policy implementation. Our findings suggest that these challenges might be addressed through clear definitions of roles of different ministries and agencies, as well as regular platforms and channels of communication to strengthen collaboration. In addition, policy awareness can be boosted by identifying, engaging and building the capacity of policy champions for domestic resource mobilisation and other health financing reforms.

**Contributors** YKO, OO, GY, KO, WM and IB were responsible for conceptualisation and design of the study. YKO and OT were responsible for writing the first draft of the manuscript. YKO, OT and IB were responsible for data collection analysis and interpretation. All authors (YKO, OO, GY, KO, WM, IB, OT, NO) were responsible for interpretation of results and critical revision of the manuscript for important intellectual content. YKO is responsible for the overall content as the guarantor and accepts full responsibility for the work and the conduct of the study, had access to the data, and controlled the decision to publish.

**Funding**  This study was funded by a grant from Bill & Melinda Gates Foundation to the Center for Policy Impact in Global Health (award/grant number: OPP1199624).

**Competing interests**  None declared.

**Patient and public involvement**  Patients and/or the public were not involved in the design, or conduct, or reporting, or dissemination plans of this research.

**Patient consent for publication**  Not applicable.

**Ethics approval**  This study involves human participants and was approved by the Duke Institutional Review Board through letter number 2019-0366. Ethical approval was also obtained from the Nigeria Health Research Ethics Committee (approval number: NHREC/01/01/2007-13/01/2020). A letter of introduction was written to all the stakeholders to be interviewed after which permission was granted for the purpose. Participants gave informed consent to participate in the study before taking part.

**Provenance and peer review**  Not commissioned; externally peer reviewed.

**Data availability statement**  Data are available upon reasonable request. The data that support the findings of this study are available on request. The data are not publicly available due to privacy and ethical restrictions.

**ORCID iDs**
Yewande Kofoworola Ogundeji http://orcid.org/0000-0002-6953-0099
Wenhui Mao http://orcid.org/0000-0001-9214-7787

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
