## [Reviewer comments · BMJ Open]

ARTICLE DETAILS

TITLE (PROVISIONAL)	Is Nigeria on course to achieve universal health coverage in the context of its epidemiological and financing transition? A knowledge, capacity and policy gap analysis (a qualitative study)
AUTHORS	Ogundeji, Yewande; Tinuoye, Oluwabambi; Bharali, Ipchita; Mao, Wenhui; Ohiri, Kelechi; Ogbuoji, Osondu; Orji, Nneka; Yamey, Gavin

VERSION 1 – REVIEW

REVIEWER	Ejemai Eboeime University of Alberta Faculty of Medicine and Dentistry, Department of Psychiatry
REVIEW RETURNED	31-May-2022

GENERAL COMMENTS	I find this article both relevant and interesting. I do have some methodological concerns though, which would need clarification. I highlight the major ones here.. kindly see the attached file for more details. 1. The authors need to ensure that the manuscript follows the Consolidated criteria for reporting qualitative research (COREQ) checklist2. The sampling approach needs to be clarified. It seems to me that the approach was snowball sampling applying the technique of maximum variation. But the way this is reported makes the sampling a little confusing. It would seem as though two independent samplings were carried out, particularly as you state initially that a small sample of cases were selected thus giving an impression of a defined finite number. Kindly clarify. Further, with the snowballing technique, when did the snowballing stop? At saturation or at resource exhaustion?3. The application of framework analysis needs to be better detailed. Framework analysis typically entails five steps: data familiarization, framework identification, indexing, charting, and mapping and interpretation. The authors need to elaborate on what analytical framework was used. The authors need to elaborate on what analytical framework was used. What aspects of the analysis were deductive, and what aspects were inductive? The reviewer provided a marked copy with additional comments. Please contact the publisher for full details.
--

REVIEWER	Martin O. C. Ota GSK Belgium
REVIEW RETURNED	14-Sep-2022

GENERAL COMMENTS	Two major concerns about this paper:  1. The title, which gives a flavor of assessment of Nigeria's readiness to achieve UHC. My concern come from the fact that very few stakeholders with unclear roles and hierarchy were involved in this study, which I consider insignificant to portray the picture the title intends to do. 2. Most of the facts in this paper are generic and applicable to most developing countries. Apart from a quote of 3% in financing there is nothing to domesticate the findings of this study to Nigeria. One would have thought that an assembly of such stakeholders should discuss with facts and figures to justify their views being applied to the country. Following the above, and to buttress the existence of knowledge gaps highlighted in the paper, the authors should state if there are documents on strategies for the "4Ds" of transition, and if these are available to all stakeholders? Furthermore, is there a national strategy and policy for achieving UHC in Nigeria? How many of the stakeholders are aware of it and its content? This will give some credibility to their views. The degree of knowledge gaps will depend on the role of the stakeholders in their various departments. These were not clearly stated. It is not clear how the investigators assessed the readiness of Nigeria to finance and drive UHC agenda from such limited and "undisclosed" stakeholder status. The donor reduction was contested by some of the study participants but authors failed to comment on this disagreement.
---

REVIEWER	Zana Wangari Kiragu Boston University School of Public Health, Department of Global Health
REVIEW RETURNED	23-Sep-2022

GENERAL COMMENTS	Thank you for this piece of work, it was extremely interesting to read, and the findings are very important to inform Nigeria's progress towards UHC. Methods: The methods of the paper are sound and very clearly described. Document review is mentioned as a means of data validation, but it is unclear which documents were reviewed; please consider briefly mentioning the data validation process. I also wonder who developed the interview guide - an iterative process is mentioned but it is not clear who developed the questions. Results: While the results address the aim of assessing knowledge, capacity and policy gaps within the context of epidemiological and funding transitions – there was minimal mention of demographic transitions. I assume this is because it did not emerge as a pertinent issue driving the knowledge/capacity/policy gaps highlighted from the data. Please consider commenting on this briefly in the discussion/conclusion, as demographic transitions are mentioned in the aim. Writing: Lastly, consider streamlining the ways participants are referred to, are they "study respondents", "stakeholders" or "key informants". The terminology gets a little bit unclear on page 14, line 16 to 18 – are all the stakeholders referred to in these phrase key informants? Citizens are mentioned as part of these stakeholders, so I assume not, as the public was not involved in this research – please consider clarifying.
---

VERSION 1 – AUTHOR RESPONSE

Reviewer 1	I find this article both relevant and interesting.	
Dr. Ejemai Eboreime, University of Alberta Faculty of Medicine and Dentistry	I do have some methodological concerns though, which would need clarification. I highlight the major ones here.. kindly see the attached file for more details (bmjopen-2022-064710_Proof_hi_Comments.pdf).(see from pt. 4)	Thank you. The comments have been addressed.
	1. The authors need to ensure that the manuscript follows the Consolidated criteria for reporting qualitative research (COREQ) checklist	This has been taken into consideration and a COREQ checklist was submitted with the paper.
	2. The sampling approach needs to be clarified. It seems to me that the approach was snowball sampling applying the technique of maximum variation. But the way this is reported makes the sampling a little confusing. It would seem as though two independent samplings were carried out, particularly as you state initially that a small sample of cases were selected thus giving an impression of a defined finite number. Kindly clarify. Further, with the snowballing technique, when did the snowballing stop? At saturation or at resource exhaustion?	Snowballing stopped once we reached the participant target. In our manuscript, we also described that we continued to interview beyond saturation for credibility. We have clarified the Sampling in the methods. See page 6, lines 13-16

	3. The application of framework analysis needs to be better detailed. Framework analysis typically entails five steps: data familiarization, framework identification, indexing, charting, and mapping and interpretation. The authors need to elaborate on what analytical framework was used. What aspects of the analysis were deductive, and what aspects were inductive?	Our approach was largely deductive (with initial themes) although we provided opportunities for emerging themes. This has been clarified in the methods. See page 6, Lines 26-29. We also provide a summarized description of our approach to the 5 steps in framework analysis. See page 6 line 27- page 8, line 6.
Additional comments in the PDF	4. If you needed to add more participants after saturation was reached for the purposes of validation, does that not suggest that you may not have been convinced that saturation had been reached?	No, this is typically good practice in qualitative research to improve the credibility and trustworthiness of the research. No new or emerging themes evolved from the additional interviews, but they added richer contexts and quotes.
	5. In line with the COREQ guidelines, it is important to state who carried out the interviews.	Interviews were completed by OT. This has been clarified on page 6 (lines 10-11)
	6. Since this is not a longitudinal study, I am unable to understand how possible it is that participants could "drop-out". It seems more likely that you intend to report that all interviewees invited accepted and participated in the study.	In some cases, participants may start the interviews and express interest in not wanting to continue with it or be a part of the study anymore. We did not have any of such cases in our study. We think it is important to highlight this.
	7. In line with the COREQ guidelines, it is important to state who carried out the data analysis.	OT, IB and YKO completed the analysis. See page 6, lines 26-27
	8. "Analysis began by reading transcripts" Who did this? How many?	OT, IB and YKO completed the analysis. See page 6, lines 26-27

	9. Details of what MDAs and states were involved would give a clearer picture about the application of maximum variation	The maximum variation we sought were within stakeholder categories working in the health sector in Nigeria. Typically, this includes government officials, academia, development partners (including civil society organization. We have included additional information on the level of the government officials. See Page 7, line 18-23
	10. "Four key themes" - These seem to be the deductive framework. "Sub-themes"- Inductive? Under each of these, an inductive process was conducted. Kindly clarify	Yes, the analysis was largely deductive. We also had an idea of the potential initial themes, but we also let the data guide eventual themes and were refined based on data. We have clarified this further. See page 6, Lines 26-29
Reviewer 2	Two major concerns about this paper:	
Martin O. C. Ota, GSK Belgium	1. The title, which gives a flavor of assessment of Nigeria's readiness to achieve UHC. My concern comes from the fact that very few stakeholders with unclear roles and hierarchy were involved in this study, which I consider insignificant to portray the picture the title intends to do.	We have provided some more details about the MDAs that were interviewed, including some indication of authority and leadership roles. We have also indicated the qualitative nature of the 'assessment' in the title. In addition, the maximum variation we sought were within stakeholder categories working in the health sector in Nigeria. Typically, this includes government officials, academia, development partners (including civil society organization. We have included additional information on the level of the government officials. See Page 7, line 18-23
	2. Most of the facts in this paper are generic and applicable to most developing countries. Apart from a quote of 3% in financing there is nothing to domesticate the findings of this study to Nigeria. One would have thought that an assembly of such stakeholders should discuss with facts and figures to justify their views being applied to the country.	We agree that there are similarities to similar contexts, which is expected. However, we do think that Participants were precise in discussing issues in the Nigerian context. This was a qualitative study that explored their knowledge, capacity, and policies on progress towards UHC within the context of ongoing health transitions. To the best of our knowledge LMICs at different stages of these transitions and they all have different health reforms they are implementing at different stages.

	Following the above, and to buttress the existence of knowledge gaps highlighted in the paper, the authors should state if there are documents on strategies for the "4Ds" of transition, and if these are available to all stakeholders? Furthermore, is there a national strategy and policy for achieving UHC in Nigeria? How many of the stakeholders are aware of it and its content? This will give some credibility to their views. The degree of knowledge gaps will depend on the role of the stakeholders in their various departments. These were not clearly stated.	The are no documents to specifically address the 4Ds. Existing policies and laws that could be considered to support the UCH agenda have been described in the manuscript. Because this is a qualitative assessment, which we have now clarified, we rely very heavily on the perceptions, opinions, and views of the respondents. While all the respondents interviewed had a good knowledge of these policies, they did express that the broader stakeholders might not have the same knowledge. They also emphasized that the implementation of these policies was slow which were described. This is further clarified on pages 12-13
	It is not clear how the investigators assessed the readiness of Nigeria to finance and drive UHC agenda from such limited and "undisclosed" stakeholder status. The donor reduction was contested by some of the study participants but authors failed to comment on this disagreement.	This study was a qualitative assessment which has now been update in the title. The assessment is not a definite one and it forms a body of evidence that can be harnessed to support progress towards UHC. In addition, the differences in donors have been clarified. For major programs such as Childhood immunizations, donor funding is reducing. However, there are other areas in which donor funding is increasing such as health systems strengthening.
Reviewer: 3	Thank you for this piece of work, it was extremely interesting to read, and the findings are very important to inform Nigeria's progress towards UHC.	
Dr. Zana Wangari Kiragu, Boston University School of	Methods: The methods of the paper are sound and very clearly described. Document review is mentioned as a means of data validation, but it is unclear which documents were reviewed; please consider briefly mentioning the data validation process.	The interview Questions were developed and modified by the authors. The documents reviewed (policy documents) have been clarified in the manuscript. Page 6, lines 2-3

Public Health	I also wonder who developed the interview guide - an iterative process is mentioned but it is not clear who developed the questions.	
	Results: While the results address the aim of assessing knowledge, capacity and policy gaps within the context of epidemiological and funding transitions – there was minimal mention of demographic transitions. I assume this is because it did not emerge as a pertinent issue driving the knowledge/capacity/policy gaps highlighted from the data. Please consider commenting on this briefly in the discussion/conclusion, as demographic transitions are mentioned in the aim.	Yes. Demographic transition did not emerge as a pertinent issue among the stakeholders interviewed compared to other health transitions. We have clarified this in the manuscript. See page 19, lines 17-21
	Writing: Lastly, consider streamlining the ways participants are referred to, are they “study respondents”, “stakeholders” or “key informants”. The terminology gets a little bit unclear on page 14, line 16 to 18 – are all the stakeholders referred to in these phrase key informants? Citizens are mentioned as part of these stakeholders, so I assume not, as the public was not involved in this research – please consider clarifying.	This is noted and the manuscript has been streamlined throughout the transcript. The lack of policy awareness was expressed by some stakeholders, and they believed that this lack of awareness cut across all stakeholder types, ranging from decision-makers to implementers, as well as citizens.

VERSION 2 – REVIEW

REVIEWER	Ejemai Eboreime University of Alberta Faculty of Medicine and Dentistry, Department of Psychiatry
REVIEW RETURNED	24-Dec-2022

GENERAL COMMENTS	All my concerns have been addressed
-------------------------------------

REVIEWER	Martin O. C. Ota GSK Belgium
REVIEW RETURNED	30-Dec-2022

GENERAL COMMENTS	Accept as it is now.
----------------------

REVIEWER	Zana Wangari Kiragu Boston University School of Public Health, Department of Global Health
REVIEW RETURNED	30-Jan-2023

GENERAL COMMENTS	Thank you for making requested revisions - this is an interesting and important paper. A note that in your responses you mention that a COREQ checklist was submitted along with the paper - this is not visible to me as a reviewer, and I hope the journal has received it accordingly.
---